



# Air mass history linked to the development of Arctic mixed-phase clouds

Rebecca J. Murray-Watson[1] and Edward Gryspeerdt[1]

[1]Department of Physics, Imperial College London, London, SW7 2BX, UK

**Correspondence:** rebecca.murray-watson17@imperial.ac.uk

**Abstract.** Clouds formed during marine cold-air outbreaks (MCAOs) exhibit a distinct transition from stratocumulus decks near the ice edge to broken cumuliform fields further downwind. The mechanisms associated with ice formation are believed to be crucial in driving this transition, yet the factors influencing such formation remain unclear. Through Lagrangian trajectories co-located with satellite data, this study investigates into the development of mixed-phase clouds using these outbreaks. Cloud
formed in MCAOs are characterized by a swift shift from liquid to ice-containing states, contrasting with non-MCAO clouds also moving off the ice edge. These mixed-phase clouds are predominantly observed at temperatures below -20°C near the ice edge. However, further into the outbreak, they become the dominant at temperatures as high as -13°C. This shift is consistent with the influence of biological ice nucleating particles (INPs), which become more prevalent as the air mass ages over the ocean. The evolution of these clouds is closely linked to the history of the air mass, especially the length of time it spends
over snow- and ice-covered surfaces, terrains may that be deficient in INPs. This connection also accounts for the observed seasonal variations in the development of Arctic clouds, both within and outside of MCAO events. The findings highlight the importance of understanding both local marine aerosol sources near the ice edge and the overarching INP distribution in the Arctic for modelling of cloud phase in the region.

## 1 Introduction

Marine cold air outbreaks (MCAOs) are formed when cold air masses from continental or sea ice sources are advected over the relatively warm ice-free ocean surface, sometimes in the wake of extratropical cyclones (Brümmer, 1999; Kolstad et al., 2009; Fletcher et al., 2016a). Due to the strong winds and the temperature contrast between the cold, dry air and the comparatively warm ocean surface, significant latent and sensible heat fluxes are generated, which can be some of the strongest on Earth (100s
- 1000 W m$^{-2}$; Shapiro et al. 1987; Renfrew and Moore 1999; Papritz and Spengler 2017; Aemisegger and Papritz 2018; Lackner et al. 2023). MCAOs are relatively common occurrences in the Arctic in all seasons except summer (Fletcher et al., 2016a). The instability caused by the turbulent heat fluxes triggers shallow roll convection and a deepening of the boundary layer, which promotes cloud formation. These clouds have distinctive mesoscale organisation and development, transforming





from overcast cloud "streets" to broken cumuliform cloud fields with increasing fetch across the ocean (Hartmann et al., 1997;
Brümmer, 1999; Pithan et al., 2018). The clouds on either side of the transition have distinct properties; in particular, the
post-transition clouds typically have a weaker shortwave cooling effect (McCoy et al., 2017; Murray-Watson et al., 2023). Due
to the frequency of these events in the Arctic, it is possible that changes to the cloud development could impact the radiative
effect of these clouds, and in turn modulate the Arctic surface energy budget. As the Arctic is changing rapidly (IPCC, 2021),
elucidating the influences on the energy budget is key to understanding the effects of climate change on the region.

Precipitation is thought to be crucial to the transition; as precipitation sublimates or evaporates, it cools the sub-cloud layer.
This creates a temperature inversion and triggers decoupling of the surface, which eventually leads to the cloud break-up (Abel
et al., 2017; Yamaguchi et al., 2017). Precipitation reduces the water content within the cloud, further contributing to the cloud's
dissipation. Therefore, processes which affect precipitation, such as aerosol concentrations, are likely to affect the transition
(Murray-Watson et al., 2023). Ice formation is hypothesized to enhance precipitation, and may accelerate the transition from
the stratocumulus to cumulus clouds in outbreaks (Tornow et al., 2021). However, the factors which affect the formation and
evolution of ice in these clouds are highly uncertain, in part due to a lack of in-situ observations in the region.

Vergara-Temprado et al. (2018) determined that the lifetime of the stratocumulus regime was influenced by the concentration
of ice-nucleating particles (INP). Prolonged stratocumulus coverage was observed under low INP conditions, while high INP
concentrations resulted in a broken cumulus clouds, leading to significant reductions in the shortwave cooling. Flight campaign
measurements of clouds in MCAOs indicate that there is typically little ice in the stratocumulus regime and substantially
more in the cumulus clouds (Abel et al., 2017; Lloyd et al., 2018). However, this shift to a more glaciated cloud regime
cannot be entirely explained by the presence of INP; Abel et al. (2017) observed a rapid enhancement of ice crystal number
concentration that was not matched by the ambient INP availability. Therefore, secondary ice production, in which an ice
crystal is formed from an existing ice crystal, might be necessary to explain the development of ice-containing clouds observed
in these outbreaks.

One of the most well-known secondary ice production processes is known as the Hallett-Mossop mechanism. This is the
process by which ice splinters are produced following the riming of a supercooled droplet onto an existing ice particle (Hallett and Mossop, 1974). These ice splinters themselves can rime onto droplets and produce more ice, therefore reducing the
dependence on INP for ice formation. The process is most efficient between -3°C and -8°C and for droplets with a diameter
greater than 24 $\mu$m or less than 13 $\mu$m (Mossop, 1978, 1985; Heymsfield and Mossop, 1984). The enhanced ice crystal concentration observed in Abel et al. (2017) occurred in lower-level cumuliform clouds which had cloud base temperatures within
the Hallett-Mossop temperature range, suggesting that this mechanism was active. Through modelling, they found that this
enhanced precipitation from this secondary ice production was essential for the stratocumulus-to-cumulus transition. Surface-based radar observations from the Cold-Air Outbreaks in the Marine Boundary Layer Experiment (COMBLE) field campaign
(Geerts et al., 2022) also found evidence of secondary ice production in MCAO clouds in the Hallett-Mossop temperature
range, with ice multiplication factors on the order of 100 (Mages et al., 2023). Although there is still uncertainty about the
physical mechanism of the Hallett-Mossop process, it is one of the only secondary ice production mechanisms parameterised
in weather and climate models.



Riming, or the freezing of supercooled droplets onto ice crystals, is also thought to influence MCAO transitions. Abel et al.
(2017) found rimed graupel particles in both the pre-and post-transition clouds. Tornow et al. (2021) suggested a mechanism called "preconditioning by riming", during which the formation of ice precipitation primes the MCAO clouds for transitioning. Through reducing the cloud water, scavenging aerosols and cooling the sub-cloud layer, increasing ice concentrations accelerated the cloud break-up.

Even given the importance of secondary ice production, it is evident that sources of INP in the region have a strong control
on MCAO cloud development. However, there is great uncertainty about INPs globally due to the diversity of sources, variety of physicochemical properties, low concentrations and seasonal cycle (Murray et al., 2012, 2021). This is particularly true in the Arctic, where INP concentration are much lower than the global average (Li et al., 2023), as sparse measurements contribute to a greater lack of knowledge (Schmale et al., 2021). However, a number of field campaigns or in-situ measurements have identified the Arctic ocean as a source of biological INP (Bigg, 1996; Bigg and Leck, 2001; Irish et al., 2017, 2019;
Creamean et al., 2019; Hartmann et al., 2021; Porter et al., 2022; Li et al., 2023). This organic matter is transported from the sea surface to the atmosphere via a bubble-bursting mechanism (Wilson et al., 2015). Wex et al. (2019) identified a seasonal cycle to INP availability around the Arctic, with higher concentrations in the summer months, likely due to enhanced biological activity spurred by increased availability of sunlight. Biological INP are distinctive as they can cause ice formation at higher temperatures than other INP such as mineral dust, and are typically the dominant INP activate above -15 °C.

Several studies have explored the influence of long-distance transport and the length of time an air parcel remains over various surface types on the INP present in those air masses. Porter et al. (2022) found that in the summertime high Arctic, the presence of high-temperature INP was associated with air parcels that had spent time over land or ocean. In contrast, time over the sea ice was associated with low aerosol concentrations (Mauritsen et al., 2011). Similarly, Irish et al. (2019) found a positive correlation between INP concentration and time over snow-free land surface and a negative correlation with time
spent over pack ice. However, they found a weakly negative relationship between INP concentrations and time spent over the open ocean. Li et al. (2023) also found low INP populations in air masses which travelled over the sea ice, and that long-range dust from Arctic sources such as Greenland was associated with particularly high INP concentrations. Bigg (1996) linked the time since the air parcel had been over the open ocean with declining INP concentrations, suggesting that the marine sources dominated INP concentrations. In contrast to many of these studies, Hartmann et al. (2021) found no connection between INP
concentrations and air mass history, and determined that the observed marine biological INP were produced locally.

MCAOs have proven a challenge to model due to the difficulties in capturing the complex system dynamics. Even higher-resolution numerical weather prediction models struggle due to a lack of knowledge around mixed-phase processes and the clouds falling into a convective "grey zone" where the necessary dynamics cannot be parameterized (Field et al., 2017). These shortcomings lead to biases between NWP modelled cloud properties and observations of cloud water path and cloud fraction
in the stratocumulus regime in MCAOs, leading discrepancies in radiative properties (Field et al., 2017). Improving the cloud microphysics scheme, in particular the ice formation processes at warmer temperatures, can lead to a better representation of MCAO clouds, but biases still exist (Field et al., 2014). Poor representation of ice processes in models causes the rapid depletion of liquid, leading to an earlier breakdown of stratocumulus clouds than observed and lower amounts of supercooled water



(Abel et al., 2017; Van Weverberg et al., 2023). In the Southern Ocean, the inability to represent supercooled water in MCAO
clouds has been identified as a central factor contributing to biases in shortwave radiation (Bodas-Salcedo et al., 2014, 2016).
In general, mixed-phase clouds are challenging to model due to the complexity of physical processes associated with their
formation and maintenance and a lack of observations of fundamental mechanisms, which lead to uncertain parameterizations
(Morrison et al., 2012). Similarly to the MCAO-specific cases, Cesana et al. (2022) found that general circulation models
(GCMs) which have more complex microphysics schemes typically produce better simulations of the liquid-to-ice ratio in
mixed-phase clouds in the Southern Ocean than simpler schemes. As mixed-phase clouds are central to the Arctic energy
budget (Shupe and Intrieri, 2004) and may prove important components of the cloud-radiative feedbacks (Tan and Storelvmo,
2019), targeting the uncertainty surrounding them is key for improving future projections of the Arctic climate.

It is evident that an advanced understanding of the formation of ice in cold-air outbreak clouds is required to improve
the modelling deficiencies. There have been several observational campaigns targeting cold-air outbreak clouds, such as the
recent COMBLE campaign in the north European Arctic (Geerts et al., 2022) and HALO-(AC)[3] flight campaign. However,
many of the observational studies are limited in space or time, and therefore provide a relatively limited picture of the cloud
development. Satellites can overcome some of these challenges, and while they do not provide as high-resolution data as the
in-situ studies, they offer the opportunity to capture the spatial and temporal dynamics of cold-air outbreak clouds on a much
larger scale. Previous studies typically have relied on snapshot satellite imagery to investigate the prevalence of ice-containing
clouds in MCAOs (e.g. Fletcher et al. 2016b), despite these properties depending on the stage of the outbreak's progression, as
ice is more likely to be found in post-transition cumuliform clouds (Abel et al., 2017).

In this work, we target the uncertainty surrounding the development of mixed-phase clouds in MCAO. In particular, we
address two aspects of mixed-phase development; the temporal scales over which the mixed-phase clouds form and the depen-
dence on temperature, and the role that air mass history may plays in the types of INP available for ice formation. To achieve
this, we use a novel method of creating a series of trajectories of air parcels moving from the sea ice and across the open ocean.
We colocate these trajectories with active remote sensing data to create a composite picture of temporal development of mixed-
phase during these outbreaks. We also consider clouds which transition from the ice edge to the ocean but are not embedded
in an outbreak to study the influences on general mixed-phase clouds in the region. To investigate the relative importance of
local aerosols versus distant INP sources, we examine the origin of the air mass and the types of surfaces it has passed over
before reaching the pack ice. We find that in most MCAOs, the proportion of mixed-phase clouds typically rapidly increases
as the cloud moves from the ice edge. These mixed-phase clouds initially form at low temperatures, but eventually become
dominant at around -13 °C, consistent with marine INP being transported to the cloud layer. We present evidence linking the
the air mass history and aerosol availability with the formation of mixed-phase clouds in both MCAO and non-MCAO clouds.
These results highlight the crucial role of INP in cloud development and the need for more detailed knowledge of their sources
in the region to be able to accurately model mixed-phase cloud development.




## 2 Methods

### 2.1 Calculating "Time Since Ice"

This work uses Lagrangian trajectories to study the temporal evolution of the cloud phase in cold air outbreaks. Detailed discussion of the trajectory generation is described in Murray-Watson et al. (2023), which in turn is adapted from Horner and Gryspeerdt (2022). In summary, hourly ERA5 (ECMWF Reanalysis v5; Hersbach et al. 2020) 1000 hPa wind data are used to advect pixels forward through time. When a pixel transitions over the sea ice edge (data from Nimbus-7 SMMR and DMSP SSM/I-SSMIS, Version 2 product; DiGirolamo et al. 2022), the pixel receives a TSI increment of 1 hour. This process is iterated for each time step, so pixels have a value of 1 added to the TSI for each hour it remains over ocean since leaving the ice edge (so pixels move from being 1 to 2 hours since ice, etc.). These TSI trajectories allow for the movement of air parcel to be tracked over the ocean after leaving the sea ice edge. When these trajectories are colocated with satellite data, they can provide insights into progression of these outbreaks. A series of reverse trajectories are also generated, measuring instead the time until an air parcel reaches the ice edge, moving towards the ice (as in Murray-Watson et al. 2023). These "Towards" trajectories allow for a comparison between the clouds moving off the ice edge and other clouds in the region, revealing the impact of any undetected sea ice on the cloud retrievals.

This work considers an extension to the trajectories method described in Murray-Watson et al. (2023) by examining the duration of time a given air parcel has spent over the sea ice and snow-covered surfaces before it leaves again and moves over the open ocean. Previous work suggests that the duration spent over the sea ice can change the air mass properties and the types of clouds formed (Silber and Shupe, 2022). In the high Arctic, the boundary layer above the pack ice can be extremely clean, with measurements of cloud condensation nuclei concentration recorded below 1 cm$^{-3}$ during the Arctic Summer Cloud Ocean Study (ASCOS) campaign (Mauritsen et al., 2011), and previous work linking air mass history with INP concentrations (Irish et al., 2019; Porter et al., 2022). The method for "time over sea ice/land" is similar to the above, but instead considers all pixels above 50N in order to better capture the air mass history, including air moving into the Arctic region. The surface type data is obtained from the Interactive Multisensor Snow and Ice Mapping System (IMS) Daily Northern Hemisphere Snow and Ice Analysis at 1km Resolution, Version 1 product (U.S. National Ice Center, 2008), and is regridded to the same polar stereographic, 25 km by 25 km grid as the sea ice data. The "Land Without Snow" and "Snow Covered Land" were used to define 'land' pixels. If a pixel contains a mix of surface types, it is categorized under the dominant type when this type occupies more than 75% of the pixel area. Otherwise, the pixel is characterised as "uncertain".

### 2.2 MCAO index

The MCAO index (M), which is an indicator of the instability of the boundary layer, is calculated as:

$$M = \theta_{skt} - \theta_{800} \tag{1}$$



with $\theta_{skt}$ and $\theta_{800}$ as the potential temperatures of the surface skin and 800 hPa, respectively (Kolstad and Bracegirdle, 2008; Fletcher et al., 2016b). The temperature data are obtained from ERA5 and processed as the wind data. An outbreak is defined as M > 0, with increasingly negative values indicative of greater boundary layer stability. MCAO are relatively common in the region and form an important part of the high-latitude weather system, particular in the non-summer months (Fletcher et al., 2016b).

## 2.3 Cloud Properties

Cloud phase data were from DARDAR v2 (raDAR/liDAR; Delanoë and Hogan 2010; Ceccaldi et al. 2013), which is produced using lidar data from CALIOP (Winker et al., 2009) and radar data from CloudSat (Stephens et al., 2008). As the radar can penetrate optically thick clouds and is sensitive to larger particles and the lidar is sensitive to thin clouds and smaller particles, this allows for a wider variety of clouds to be studied than if the datasets were used in isolation. As the lidar data is produced at a higher resolution than CloudSat, several CALIOP retrievals are matched to each radar retrieval when merged, resulting in a vertical resolution of 60 m and a horizontal resolution of 1.1 km. The phase determination algorithm incorporates data from the CALIOP L1B attenuated backscatter profiles, the CALIOP L2 Vertical Feature Mask and the CloudSat 2B-GEOPROF radar and ECMWF-AUX temperature data. Cases where the cloud top height below 720 m are excluded due to surface clutter affecting the radar retrievals, as in other studies (e.g. Kay and Gettelman 2009). The phase fraction for each profile is calculated as the number of DARDAR retrievals for each flag divided by the number of DARDAR retrievals for each 25 km by 25 km TSI grid box. For this study, the 'ice' category comprises DARDAR phase flags 'Ice', 'Spherical_or_2D_ice' and 'Highly_concentrated_ice' and DARDAR flag 'Supercooled_and_ice' is considered 'mixed phase'. We adopt the method of defining mixed-phase profiles in DARDAR from Danker et al. (2022); if a 'mixed' layer is above an 'ice' layer, a 'supercooled' above 'ice', any combination of 'supercooled' and 'mixed' or any combination of 'supercooled' or 'mixed' above 'ice'. The 'warm liquid' and 'supercooled liquid' DARDAR categories are combined to create a single 'liquid' category. Multi-layer clouds are excluded by removing profiles which have more than three 'clear sky' or failed retrievals (equating to a distance of 240 m) between cloud layers, as in previous work (Danker et al., 2022). Clouds with cloud top heights recorded above 5000 m are excluded to limit the inclusion of high-level clouds not associated with MCAOs. Although this may exclude some MCAO clouds, other work has found clouds tops are typically much lower than this (Fletcher et al., 2016b; Lackner et al., 2023; Mateling et al., 2023).

Cloud effective radius ($r_e$) measurements were obtained from the Moderate Resolution Imaging Spectroradiometer (MODIS) Level 2 Collection 6.1 data product (MYD06_L2; Platnick et al. 2017), regridding into a 25 km by 25 km polar stereographic grid. Following Grosvenor and Wood (2014), pixels with high solar zenith angles (>65°) and high viewing angles (>50°) were removed due to potential biases in $r_e$. The dataset was filtered to include only single-layer, liquid clouds using "Cloud_Multi_Layer_Flag" and "Cloud_Phase_Optical_Properties". The cloud top phase was determined using the cloud optical properties retrieval. Only pixels with a low heterogeneity index were included ("Cloud_Mask_SPI" < 30; Zhang and Platnick 2011).





# 3 Results

## 3.1 Phase evolution as a function of time since the ice edge

Clouds created in MCAOs have a distinctive phase evolution relative to other clouds in the region (Figure 1), most notably from the ice- and mixed-phases. MCAO clouds are characterised by the rapid decline of the liquid phase down to about 20% after 6 hours, with a corresponding increase in mixed-phase clouds. While the liquid fraction then remains relatively steady, the mixed-phase proceeds to glaciate and convert to ice-only clouds. In contrast, for both non-MCAO and Towards clouds, the liquid phase remains dominant. The phase fractions remain relatively steady as non-MCAO clouds move off the ice edge, with nearly twice as much liquid cloud as mixed-phase and very few (about 3%) ice clouds. Although the composition of Towards clouds also shifts towards ice-containing phases as they move further north, the shift is very gradual, and liquid clouds are still dominant. This shift to mixed-phase is likely in part due to decreases in cloud temperature as the clouds move towards the ice. Additionally, as these clouds move north from lower latitudes, the air masses may potentially have more INP than the air which has spent greater lengths of time over the sea ice (discussed more in Section 3.3), and therefore more readily form mixed-phase clouds. Broadly, the dominance of liquid-containing phases in the Towards clouds agrees with previous climatologies of low-level Arctic clouds (e.g. Shupe et al. 2006; Cesana et al. 2012; Yan et al. 2020), but does show more liquid-only clouds than some other studies (e.g. Shupe 2011), although this may be due to different study regions. Using DARDAR, Fletcher et al. (2016b) also determined that clouds within outbreaks had higher ice fractions compared to other regional clouds.

This shift in MCAO clouds towards more glaciated state is in agreement with previous observations; Abel et al. (2017) found much larger quantities of ice in the post-transition cumulus clouds relative to the upstream stratocumulus deck. The following sections will explore the potential roles of the air mass history and the cloud top temperature in influencing the observed disparities in ice-containing phases between MCAO clouds and non-MCAO trajectories.

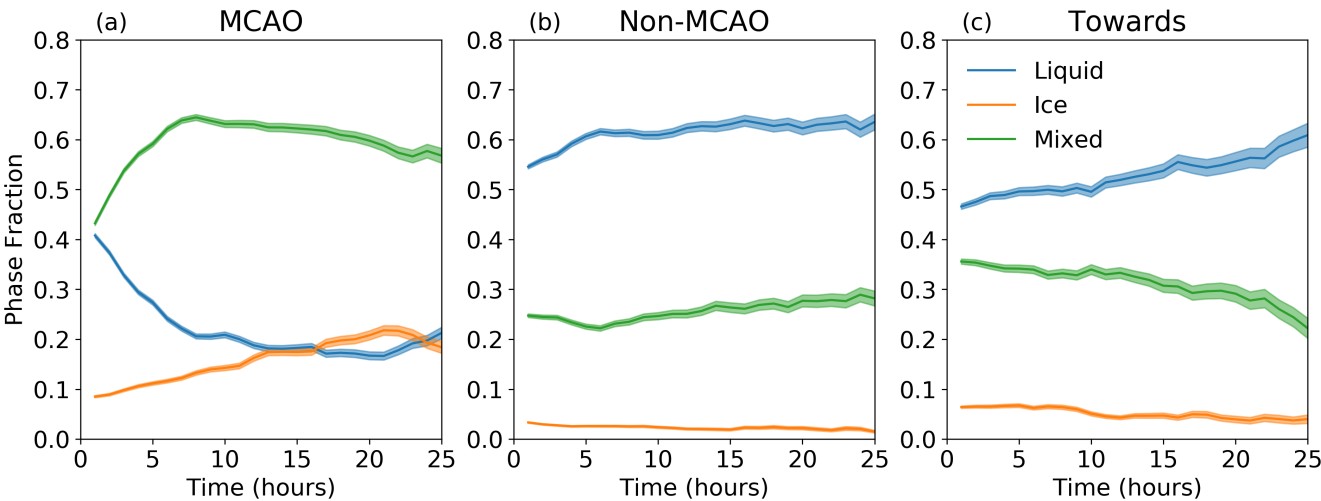

**Figure 1.** Evolution of cloud phase as a function of time for (a) MCAO, (b) non-MCAO and (c) Towards trajectories.



## 3.2 Mixed-phase development as a function of temperature and time since ice

Cloud top temperature is an important factor in the formation of ice in clouds (e.g. Korolev et al. 2003; Shupe et al. 2006; Westbrook and Illingworth 2011; de Boer et al. 2011), typically with increasing amounts of ice at lower temperatures. However, supercooled liquid clouds have also been observed to exist at temperatures below -15°C (e.g. D'Alessandro et al. 2021; Danker et al. 2022; Carlsen and David 2022). Figure 2 shows evolution of the mixed-phase fraction for MCAO and non-MCAO clouds as a function of the time since ice and cloud-top temperature. Mixed-phase clouds are initially only dominant at very low

temperatures (-23°C) in MCAO. As the clouds progress from the ice edge, the temperature at which the majority of clouds are mixed-phase increases, until a plateau at approximately -13°C after 7 hours. The low mixed-phase fraction at very low temperatures further into the trajectories (bottom-right of Figure 2 (a)) is due to the shift to ice-only clouds (Figure S1).

This is in contrast to the non-MCAO clouds; from the point of leaving the ice edge onwards, mixed-phase clouds are prevalent below -13°C and supercooled liquid dominant at warmer temperatures. Figure 2 (c) underlines the differences in this

initial development by emphasising the initial relative lack of mixed-phase clouds early in MCAO (the gradient highlighted as Region A). This gradient in the temperature at which mixed-phase clouds become the majority has previously been observed by Carlsen and David (2022), who found that the temperature at which a cloud regime switched from being liquid-dominant to mixed-phase dominant (which they called T*) increased as a function of distance from the ice edge, until a plateau is reached at about -15°C. They also observed a seasonal dependence; during the summer, there was a slightly weaker gradient in T* from

the ice edge (from about -17°C to -15°C) than in the winter (from about -22°C to about -15°C). This may be due to the seasonal differences in MCAO events; there are relatively few MCAO in summer (Figure 3), so they may have been observing events more similar to Figure 2 (b), i.e events which are moving off of the ice edge, but are not necessarily MCAO. Conversely, in winter, their observations match Figure 2 (a) more closely. The seasonal dependence of these MCAO and non-MCAO cloud evolution is discussed further in Section 3.3.

Figure 2 (a) also reveals another interesting feature of the MCAO clouds; after about 7 hours, a significant proportion of the clouds are mixed-phase at higher temperatures (between -13°C and -3°C). On average, the mixed-phase proportion at these temperatures in MCAOs is 15-20% greater than in non-MCAO clouds (Figure 2 (c), highlighted as Region B). The potential cause of the development of these higher-temperature mixed-phase clouds in outbreaks are discussed in Section 4.

## 3.3 Seasonal variability in mixed-phase development

Previous work has found a strong seasonal cycle in MCAO, with a maximum frequency of occurrence in the winter and minimum in the summer months (Fletcher et al., 2016b). Figure 3, which accounts for the MCAO index just as the air parcel leaves the ice edge, shows the number of MCAO pixels in the northern hemisphere winter in the region of study are orders of magnitude less common than other seasons for the period of study.

Figure 4 shows that for most seasons, the pattern of mixed-phase cloud development in MCAO is similar to the annual

aggregate; initially, mixed-phase clouds are only dominant at very low temperatures, and become increasingly dominant at



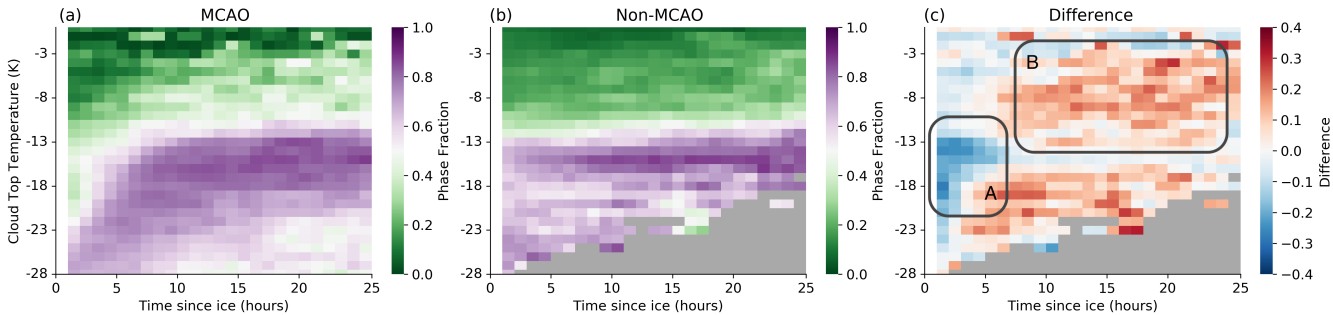

**Figure 2.** Evolution of mixed-phase cloud fractions as a function of time and cloud top temperature for (a) MCAO, (b) non-MCAO. (c) is the difference between them. Grid points with fewer than 500 successful retrievals are coloured in grey. In plot (c), red indicates that there are more mixed-phase clouds in MCAOs for that given cloud top temperature and TSI than in non-MCAOs, and blue indicates the opposite.

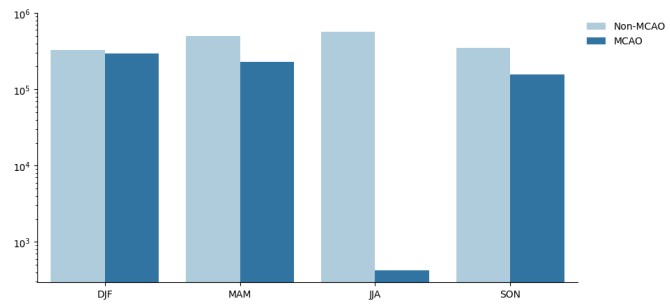

**Figure 3.** Number of MCAO and non-MCAO pixels recorded at TSI = 1 for each season.

higher temperatures until a plateau at around -13°C, although SON also more mixed-phase clouds at higher temperatures later in the outbreak. JJA is an anomaly; its behaviour is very similar to the annual non-MCAO cloud development (Figure 2 (b)).

In contrast, Figure 5 shows the seasonal development of clouds in non-MCAO events. This time, MAM, SON and JJA replicate the annual pattern shown in Figure 2 (b). However, the non-MCAO DJF mixed-phase cloud development more
strongly resembles an MCAO event (Figure 2 (a)). A potential cause for seasonal variation in both MCAO and non-MCAO mixed-phase development is discussed in Section 3.4.1.

### 3.4   Effects of air mass history

Initially focusing on the annually-averaged general cases, there are several possible causes of the distinctive mixed-phase evolution in the MCAO clouds seen in Region A of Figure 2 (c). One may be due to the air mass history. Figure 6 (a) shows
the duration of time that air parcels which form MCAO and non-MCAO events typically spend moving over the sea ice before crossing over to open ocean. The pack ice is typically a poor source of INPs, with low INP concentrations being associated with longer times over ice (Porter et al., 2022; Li et al., 2023). Although AOD is an imperfect proxy for INP, especially in a



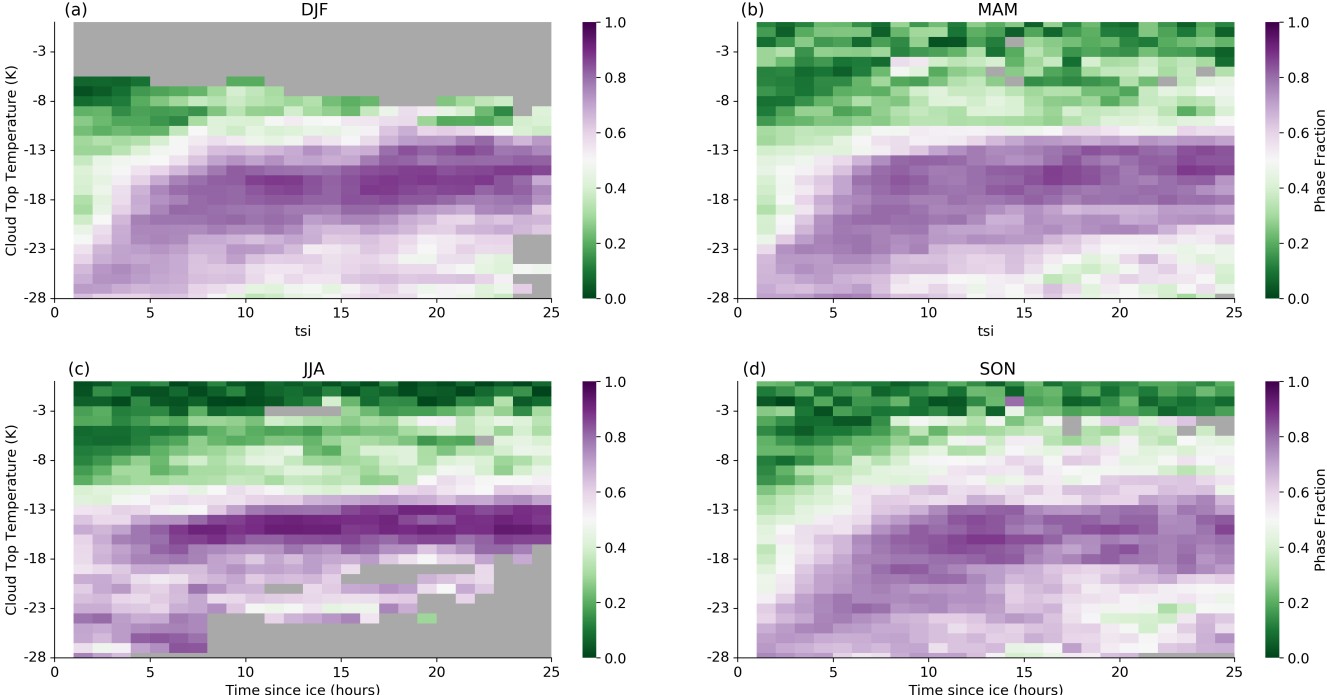

**Figure 4.** Evolution of cloud phase as a function of time and cloud-top temperature (from the DARDAR product) for MCAO clouds in (a) December, January and February, (b) March, April and May, (c) June, July and August and (d) September, October, November. Points for which there were fewer than 500 retrievals are coloured grey.

region as poorly constrained as the Arctic, Copernicus Atmosphere Monitoring Services AOD reanalysis data (Inness et al., 2019) was used to indicate the cleanliness of the air masses just as they move off of the ice edge (at a TSI = 1). Figure 6 (b)
indicates that these MCAO air masses typically have lower AODs than non-MCAO air masses. Therefore, as they move off the ice edge, INP may be locally sourced from the ocean surface and transported to the cloud layer, and then activated to form ice particles. The higher wind speeds typically observed in MCAOs (Kolstad, 2017) may aid the transport of these INP via the bubble-bursting mechanism (Wilson et al., 2015), which has previously been observed (Inoue et al., 2021). The temperature of the liquid/mixed-phase switch (approximately -13°C after 7 hours in MCAO) is often considered the temperature at which
biological INP tend to dominate over other potential INP, such as mineral dust (Murray et al., 2012). Therefore, it appears that while biological INP may already be present in the non-MCAO clouds as they move over the ice edge (indicated by the presence of mixed-phase clouds form at -13°C close to the ice edge), they are absent from the MCAO air masses, and must be from a local source.



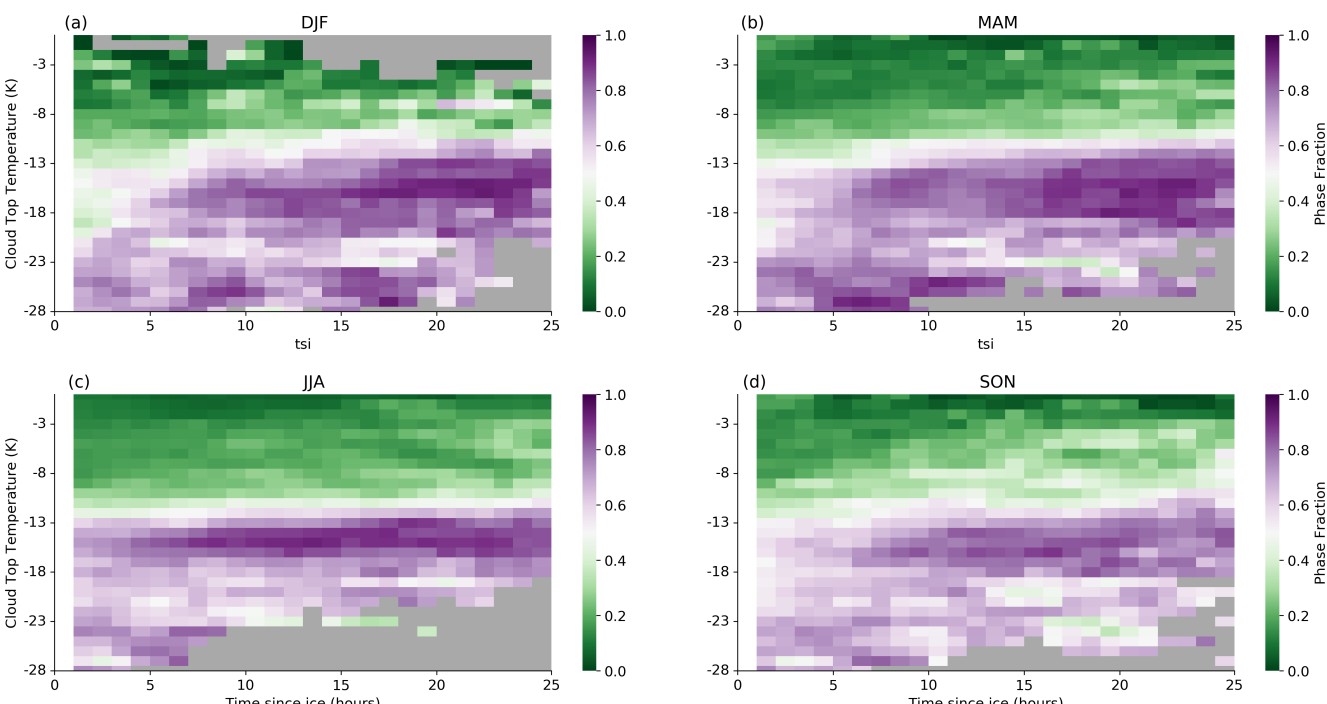

**Figure 5.** As Figure 4, but for non-MCAO clouds.

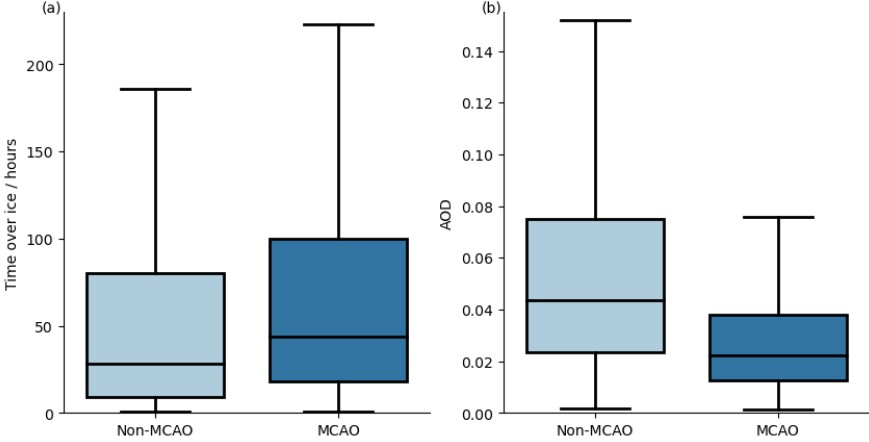

**Figure 6.** (a) Time spent over ice before leaving the ice edge and (b) AOD for MCAO and non-MCAO trajecotries, just the air leaves the ice edge (at TSI = 1).



### 3.4.1 Seasonal dependence of air mass history

What causes the non-MCAO DJF clouds to develop like most MCAO clouds and MCAO JJA clouds to develop like most non-MCAO clouds? One possible explanation is that the DJF non-MCAOs are very clean, therefore causing longer times for the mixed-phase clouds to form. Similarly, we may expect the JJA MCAOs to have higher aerosol loads, therefore allowing the mixed-phase to rapidly develop. In this case, we would expect the time spent over ice to be important, due to the potentially cleaner air masses with longer times over ice. However, Figure 7 (a) shows that both DJF cases generally spend less time over

ice than the other cases, contrary to what would be expected. Additionally, with the JJA MCAOs spending comparable amounts of time over ice to MAM and SON cases (although, as there are fewer MCAO cases during the summer, this may be due to noise). While the CAMS AOD data (Figure 7 (b)) shows that DJF MCAO and non-MCAO are both relatively clean, the SON non-MCAOs and JJA MCAOs also have similar AOD. This may be due to CAMS not capturing the types of aerosols present in SON and JJA, which are expected to be dominated by local biological sources which are absent in the DJF case.

Looking further back into the air mass history provides some potential answers. Figure 8 shows the average time that the air parcels spent over snow-covered land before then moving onto ice for each season. Both non-MCAO and MCAO cases in winter typically spend long times over snow-covered land before reaching the ice, while the JJA cases typically spend very little, as expected when considering the seasonal cycle of surface cover (Figure S2). For both MAM and SON, the MCAOs usually spend a longer time over snow than non-MCAOs. Carlsen and David (2022) suggested that snow-covered surfaces

prevented biological INP release and therefore suppressed mixed-phase formation at higher temperatures. Therefore, it may be that in summer, terrestrial INP are collected from the surrounding land mass and is available, along with local, marine biological INP, for ice formation. Terrestrial INP have previously been found to be an important source of higher-temperature INP in the summertime Arctic (Pereira Freitas et al., 2023). In contrast, in winter, no such INP are available for either the MCAO or non-MCAO events when the air is advected over land, and therefore it takes time for the local INP to be transported

from the ocean surface to the cloud layer in both cases. Although marine biological INP are scarcer in the winter, they are not entirely absent (e.g, Creamean et al. 2022; Hartmann et al. 2020; Pereira Freitas et al. 2023).

To test the idea that the time spent over ice- and snow-covered surfaces impacts the temperature at which mixed-phase clouds initially form through INP availability, Figure 9 shows the development of MCAO clouds for DJF in cases where the air parcel has spent either more or less than two days over these INP-limiting surfaces. As is highlighted on Figure 9 (c) Region

C, those cases which spend less time over sea ice form ice at higher temperatures within the first few hours over ocean. In contrast, the cases which have spent longer times over ice are typical of the general MCAO cases, suggesting that they are devoid of higher-temperature INP as they move off the ice edge and these aerosols must be sourced from the ocean surface. Additionally, the difference in phase development is pronounced only at the early stages of the trajectories. At later times and lower temperatures (Figure 9 (c) Region D), the phase compositions are very similar. This further suggests that air mass history

primarily affects the formation of mixed-phase clouds, due to their reliance on INP availability.

In summary, for the initial development of ice as the cloud moves from the ice edge, the air mass history appears to influence ice formation in both the MCAO and non-MCAO cases. MCAO air masses typically spend more time over ice- or snow-



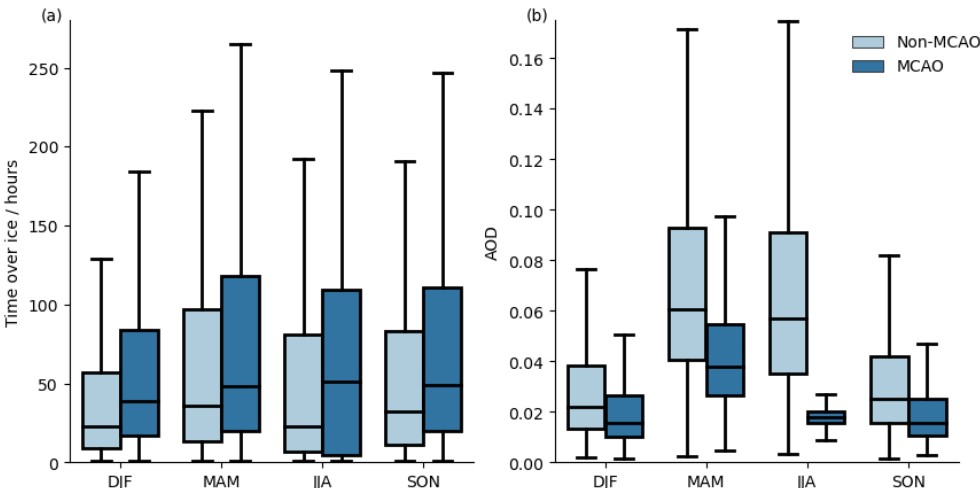

**Figure 7.** As Figure 6, but broken into seasons.

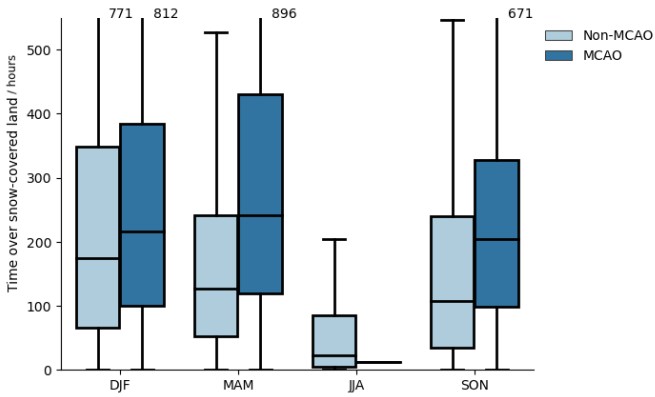

**Figure 8.** As Figure 7, but for time spent over snow-covered land before moving onto the ice edge.

covered surfaces. As these surfaces may suppress the transport of INP to the atmosphere (e.g. Carlsen and David (2022)), they may be relatively clean by the time they traverse the ice-ocean boundary. As the air moves over the ocean, the turbulent fluxes and strong wind allow for the transport of INP to the cloud layer, allowing marine INP to form ice. In contrast, non-MCAO typically spend less time over these INP-blocking surfaces, allowing for non-local INP to be carried over the ice edge, allowing for earlier mixed-phase formation. There are two exceptions; in winter, the surrounding land is covered in snow and ice, so terrestrial INP aren't available, and there is only a weak source of marine biological INP. Additionally, the longer time spent travelling over these surfaces means that any existing INP in the air mass is more likely to be removed without replenishment. Therefore, the wintertime non-MCAO clouds develop very similarly to the MCAO clouds. However, if the DJF air masses spend relatively little time over ice, mixed-phase clouds do form at around -13°C close to the ice edge due to the availability





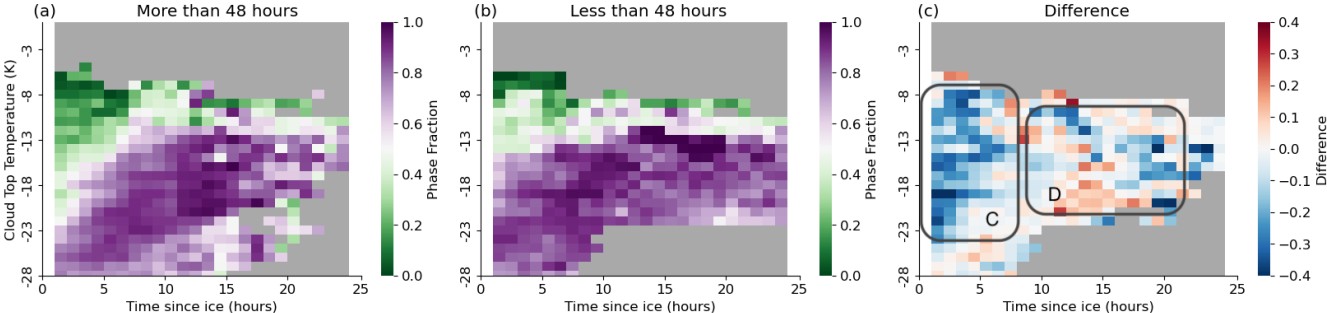

**Figure 9.** As Figure 2, but for DJF MCAO cases that have spent (a) more than 48 hours and (b) less than 48 hours over snow- and ice-covered surfaces. (c) is the difference between (a) and (b). Grid points with fewer than 250 successful retrievals are coloured in grey. Blue indicates that the cases which have spent longer times over ice have lower mixed-phase cloud fractions.

INP. In contrast, abundant terrestrial and biological INP in summer allow for the ready formation of ice in the MCAO. In spring and autumn, although non-MCAO clouds do travel over ice-covered surfaces, the biological sources are still active enough to allow for quick ice formation as the clouds move from the ice edge.

## 4 Potential causes of the higher-temperature mixed-phase clouds

The prevalence of mixed-phase clouds with CTT warmer than -10°C (Region B in 2 (c)) is challenging to explain with the data used here. There may be several causes these higher-temperature mixed-phase clouds, namely: secondary ice production, the presence of INP active at these high temperatures and potential retrieval biases, which are discussed below.

### 4.1 Secondary ice production

Secondary ice production has previously been cited as a cause of mixed-phase development in MCAO; during a field campaign, Abel et al. (2017) found that the number of ice crystals in the post-transition cumuliform clouds was several order of magnitudes greater than what would be expected given the ambient INP concentrations. Mages et al. (2023) also found evidence of secondary ice production in the later stages of an outbreak. In both cases, cloud temperatures were within that of the Hallett-Mossop mechanism and strong updrafts were recorded. The presence of moderate to strong updrafts has previously been observed to be important to the initiation of secondary ice production (Sullivan et al., 2018; Korolev et al., 2020; Luke et al., 2021). As non-MCAO clouds are not subject to the same instability as MCAO clouds, and therefore do not have as strong updrafts, this may explain the absence of the mixed-phase at higher temperatures. Although the mixed-phase enhancement can be seen at lower temperatures than what is usual for the Hallett-Mossop temperature range (down to about -12°C in Figure 2 (c)), this may be due to the fact that the cloud top temperatures are used; the cloud base would be warmer than this, and therefore potentially in the appropriate temperature range. While temperatures up to -3°C may appear relatively warm for clouds associated with MCAOs, this may be due to the satellite observing convective clouds at different stages of development,





as some clouds transition through these warmer temperatures until they reach their final cloud top temperature (Lensky and Rosenfeld, 2006).

Secondary ice production is thought to be more efficient in the presence of larger liquid droplets (Rangno and Hobbs, 2001; Rosenfeld et al., 2011; Luke et al., 2021). To investigate this, the the trajectories in Figure 2 (a) were divided into high and low $r_e$, based on $r_e$ retrievals taken within 6 hours of the DARDAR retrieval used in Figure 2. The $r_e$ data were obtained from MODIS, and due to the uncertainties associated with retrieving $r_e$ in mixed-phase clouds (Khanal and Wang, 2018), liquid-only retrievals were used. The aim is to consider the size of liquid droplets in the cloud before they transition to a mixed-phase regime. This excludes many cases for which a liquid-only MODIS retrieval could not be made (particularly in during polar night, as MODIS requires reflected sunlight to retrieve $r_e$). A threshold of 14 $\mu$m was used to demarcate high and low $r_e$. The results are shown in Figure 10.

Due to the seasonal bias in MODIS retrievals, the general mixed-phase development in Figure 10 most closely resembles the JJA MCAO (which, in turn, resembles the non-MCAO clouds). Therefore, it is uncertain how representative these results are. However, it can be seen from Figure 10 (c) that clouds which had larger droplets earlier in the development did go on to produce more mixed-phase clouds at the higher temperatures observed in Figure 2. The enhancement exists across the temporal development, although it is stronger after about 7 hours. Although not conclusive, this supports the argument that secondary ice production gives rise to these higher-temperature mixed-phase clouds.

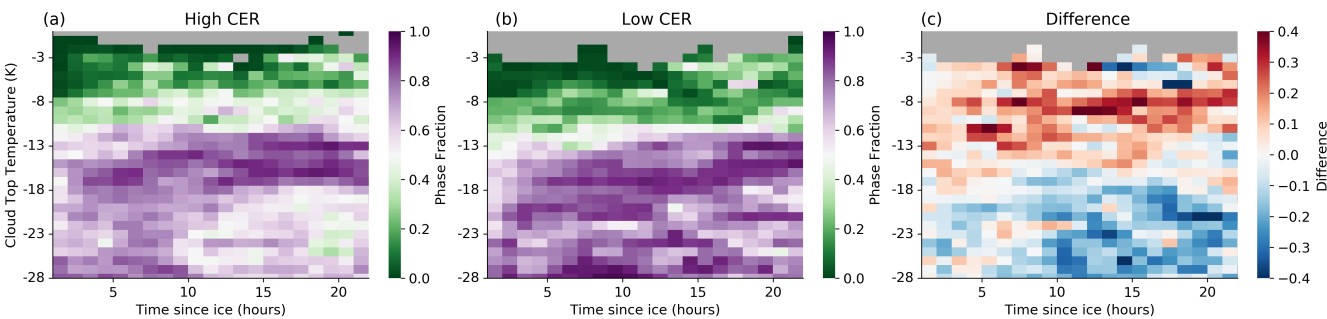

**Figure 10.** Evolution of mixed-phase cloud fraction as a function of time at cloud top temperature for clouds in MCAO events that have successful MODIS $r_e$ retrievals for the liquid phase (a) above 14 $\mu$m, (b) below 14 $\mu$m. $r_e$ retrievals must be at most 6 hours before the DARDAR retrieval to be included. (c) shows the difference between (a) and (b). Grid points with fewer than 250 successful retrievals are coloured in grey. Note that due to data limitations this threshold is lower than for Figure 2. Red indicates that clouds which have larger droplet earlier in the trajectory go on to have higher mixed-phase fractions than clouds which had smaller droplets.

## 4.2 Higher temperature INP

Another possible explanation is the activation of INP at these higher temperatures (between -10°C and -3°C). Biological INP in the seas around the ice pack have been recorded as being active as warm as -5°C (McCluskey et al., 2018; Irish et al., 2019; Wex et al., 2019; Li et al., 2023); albeit at low concentrations. Due to the scarcity of INP which would nucleate ice, it may take



longer for them to activate, leading to the delay in the formation of these higher-temperature mixed-phase clouds. However, as these higher-temperature INP are biological, a strong seasonal cycle is expected; although Figure 4 shows some seasonal differences, there is not a notable peak in summer, where biological activity is highest. Furthermore, it does not explain why

this phenomenon is seen in MCAO but not non-MCAO clouds, unless the enhanced updrafts and instability in MCAOs greatly increased the transport of these INP to the cloud layer, evidence of which has been observed previously (Inoue et al., 2021). More measurements of INP around MCAO would be required to determine the contribution of high-temperature biological INP to mixed-phase cloud formation.

## 4.3 Retrieval biases

It is possible that the higher-temperature mixed-phase clouds are not a real effect but appear due to the radar misclassifying supercooled drizzle as ice, particularly above -10°C, due to their larger size (Zhang et al., 2017, 2018). As noted by Danker et al. (2022), this may be a particular issue with DARDAR when the lidar is attenuated in supercooled-topped clouds, and heavier precipitation would be required for a CloudSat detection (Marchand et al., 2008). Drizzle has previously been observed to form in MCAOs (Abel et al., 2017; Lloyd et al., 2018), and supercooled drizzle in general has been recorded at temperatures

down to -25°C (Silber et al., 2019). Additionally, droplets in liquid clouds formed during MCAOs grow to precipitation-size (about 15 $\mu$m; Rosenfeld and Gutman 1994), whereas those not in outbreaks do not (Murray-Watson et al., 2023), potentially explaining the differences between Figure 2 (a) and (b). Schirmacher et al. (2023) found that CloudSat may also underestimate precipitation, particularly frozen precipitation, in outbreaks due to limitations such as surface clutter creating a blind zone for the radar close to the surface. While the COMBLES campaign took in-situ precipitation measurements during outbreaks, and

found that the cumuliform clouds were typically precipitating (Mages et al., 2023), with frozen precipitation being common, there is still a great deal of uncertainty about the phase of the precipitation created in these events. Therefore, it is difficult to assess how prevalent retrieval errors may be. More in-situ measurements of the the frequency and characteristics of supercooled drizzle would be required for a more robust understanding of precipitation in these events and to target the retrieval uncertainty.

With the data used in this work, identifying the cause of mixed-phase clouds forming above -10°C late in the MCAO events is challenging. Some evidence such as the influence of droplet size and the absence of similar formations in non-MCAO events suggests secondary ice production, which is consistent with in-situ observations of MCAO clouds. However, we cannot conclusively rule out other potential causes of the appearance of Region B in Figure 2. Other factors, such as snow lofted from the surface nucleating ice in the cloud (Geerts et al., 2015), cannot be ruled out with the current analysis. This may be

especially relevant in MCAO due to the high wind speeds, turbulence and the presence of frozen precipitation (Mateling et al., 2023). Satellite data alone is unlikely to be able to determine the formation mechanism of these late-stage mixed-phase clouds, highlighting the need for more detailed observations of this phenomenon.



## 5 Discussion

The results presented here are consistent with the air mass history influencing the development of mixed-phase clouds, both
in MCAO and non-MCAO events. However, this trajectory analysis does not allow for the precise origin of the air masses to
be determined, and therefore there are uncertainties in linking these trajectories to potential INP sources in the wider Arctic
region. Additionally, there may be potential INP sources in the pack ice, such as open leads, which have previously been cited
as aerosol sources at high latitudes (Bigg and Leck, 2001). These features may be on the order of several hundred meters in
size (Wadhams et al., 1985; Li et al., 2020), and therefore below the resolution of the sea ice product (Cavalieri et al., 1996).
However, Porter et al. (2022) found that open leads were very weak sources of INP, and that air that had spent time over
the pack ice typically had very low aerosol concentrations. Hartmann et al. (2020) found that INP from leads were active at
high temperatures and suggested that they were from a biogenic source. Therefore, if INP from features like open leads were
prevalent in MCAOs, we may expect more mixed-phase cloud formation at higher temperatures as soon as the air moves off
of the ice edge (low TSI) in Figure 2 (a). As open leads cover a higher fraction of the sea ice area in summer than winter
(Lindsay and Rothrock, 1995; Wang et al., 2016), these features may be partially responsible for the similarity between the
MCAO and non-MCAO JJA cases. However, there is also a high fractional coverage of leads in the autumn, and yet the MCAO
and non-MCAO development remain distinct. As there is disagreement regarding the efficacy of leads acting as INP sources,
more measurements would be required to determine their influence on cloud phase processes in the Arctic region. Whether it
be through sources such as leads or INP transported from adjacent snow-free land, these results still indicate the importance of
air mass history in the formation of the mixed-phase clouds.

It should also be noted that the analysis here is based on the cloud top temperature, and the DARDAR data product derives
cloud top temperature from ECMWF auxiliary data (ECMWF-AUX; Partain 2022). This may introduce potential uncertainties.
Furthermore, DARDAR phase classification algorithm also uses the ECMWF wet bulb temperature to distinguish between ice
(below 0°C) and liquid or rain (above 0°C), which may reduce the phase classification accuracy at higher temperatures (such
the mixed-phase clouds seen in Figure 2 (a)). However, previous work has found good agreement between the cloud top
temperature when using ECMWF-AUX temperatures and in-situ radiosonde data (e.g. McErlich et al. 2021) or other retrieved
satellite data (e.g. Christensen et al. 2016), so the effects of potential biases for these may be limited.

Mixed-phase clouds are of interest due to the effect of ice on the lifetime of the stratocumulus clouds, with higher INP
concentrations hastening the breakdown of the high-albedo stratocumulus deck through precipitation-related mechanisms
(Vergara-Temprado et al., 2018; Murray et al., 2021; Tornow et al., 2021). However, due to limitations with CloudSat in
MCAOs (Schirmacher et al., 2023), DARDAR struggles to detect precipitation in these events. Therefore, we are unable to
determine the effects of the formation of ice on the precipitation frequency in these clouds. Additionally, as MODIS retrievals
struggle with mixed-phase clouds, which are often topped by a supercooled liquid layer and therefore may be retrieved as
liquid-only clouds. An analysis on the effects of ice on cloud properties and lifetime is also beyond the scope of this research.
To achieve this, high-resolution measurements, both spatially and temporally, of ice formation and the effects on properties
such as cloud fraction and water path would be required.





## 6    Conclusions and Outlook

Marine cold-air outbreaks are important components of the Arctic weather system. As the cold air mass flows over the ice edge over the relatively warm open ocean, intense turbulent heat and surface fluxes promote cloud formation. These clouds undergo

a distinctive evolution from high coverage stratiform decks to broken cumuliform cloud fields. These post-transition clouds typically have a much lower shortwave cooling effect due to their lower cloud fractions. Therefore, factors which may change the timing of the transition can change the role these clouds play in the Arctic energy budget. In particular, the presence of ice or frozen precipitation is thought to be key to the cloud evolution. However, the conditions influencing ice formation in these events are uncertain. Here, we have developed a method to observe the evolution of the cloud phase over time in these

outbreaks, and considered the role of air mass history on the cloud evolution.

After leaving the ice edge, MCAO clouds exhibit a rapid decline in the liquid phase, with a corresponding increase in mixed-phase clouds, and some amount of glaciation into ice-only clouds (Figure 1). In contrast, the liquid phase is persistently dominant in non-MCAO clouds, with particularly few ice-only clouds. The shift towards a more glaciated state in MCAO clouds is in agreement with previous in-situ and measurement campaign observations (Abel et al., 2017; Lloyd et al., 2018;

Mages et al., 2023), with higher amounts of ice found in the post-transition cumuliform clouds than in the original stratiform decks.

Mixed-phase clouds predominantly exist at low temperatures initially (around -20°C), gradually shifting to form at higher temperatures of around -13°C (Figure 2). In contrast, mixed-phase clouds form at higher temperatures in non-MCAO very close to the ice edge. There is a seasonal dependence to the formation of ice in MCAOs and non-MCAOs (Figures 4 and 5), with

wintertime non-MCAO clouds only forming mixed-phase at low temperatures initially and JJA MCAO forming ice at -13 °C at the ice edge. These differences in evolution appear to be related to air mass history; air masses which form MCAO typically spend longer times over ice- and snow-covered surfaces, which usually are poor sources of INP (Figures 7 and 8). Therefore, the time taken to form mixed-phase clouds at higher temperatures appears to the associated to the transport of local, marine biological INP to the cloud layer. In contrast, non-MCAO events typically spend less time over snow and ice, and therefore may

carry more INP. Although the CAMS AOD data somewhat corroborates the results, with MCAO air parcels typically being cleaner, the difficulties in representing biological sources and INP in models limits the utility of the reanalysis data. Despite several measurement campaigns, there is still considerable uncertainty about aerosol sources and sinks in the Arctic (Schmale et al., 2021). The results presented here highlight the need for more accurate measurements of the sources of biological INP in the region, particularly close to the ice edge, and the mechanisms of transport to the cloud layer. Further knowledge of aerosol

transportation pathways around the Arctic is also required, especially for understanding the seasonal differences in aerosol sources.

Further into the MCAO development, there is some evidence of mixed-phase clouds occurring with cloud top temperatures above -10°C (Figure 2). Although evidence of secondary ice production has previously been observed in MCAO clouds, particularly in regions of strong updrafts or high supercooled fraction, (Abel et al., 2017; Mages et al., 2023), it is difficult

to determine decisively with satellite data if the mechanism is at play here. Karalis et al. (2022) showed that secondary ice



formation was important for the development and break-up of MCAO clouds, particularly mechanisms such drop-shattering and ice-ice collisional break-up. However, parameterizations of these mechanisms are highly uncertain and not widely implemented in climate models. Further laboratory and in-situ studies are required to understand these mechanisms, which can improve the representation in the models and in turn help to understand whether or not secondary ice production contributes to the higher-

temperature mixed-phase clouds observed in MCAO.

This study provides evidence consistent with previous modelling work showing that MCAO cloud development is sensitive to INP concentration (Vergara-Temprado et al., 2018). Future changes in Arctic INP concentrations are expected to change the radiative properties of the clouds. Increasing temperatures mean that the Arctic is increasingly snow- and ice-free, therefore potentially making more INP sources more available for greater parts of the year (Ardyna et al., 2014). These increases in INP

may further increase proportion of mixed-phase clouds in MCAOs, or help ice form earlier into the outbreak, and cause these clouds to break up more rapidly (Tornow et al., 2021). During polar day, both of these changes would decrease the cooling effects of the clouds, leading to a positive feedback. However, poor parameterizations of INP in climate models, along with difficulties of modelling mixed-phase cloud properties and the dynamics of a complicated system like a MCAO, mean that understanding the impact of changes in aerosol concentrations is challenging. To accurately estimate the climate impact of this

effect on a large scale, it's necessary to conduct detailed measurements of both aerosols and mixed-phase clouds across the Arctic.

*Data availability.* The MODIS data were obtained from NASA Goddard Space Flight Center (https://modis.gsfc.nasa.gov/data/; Platnick et al. 2017, last accessed July 2022). The ERA5 data were obtained from the Climate Data Store (https://cds.climate.copernicus.eu/cdsapp#!/dataset/reanalysis-era5-complete; Hersbach et al. 2020, last accessed June 2022), and the CAMS reanalysis from the Atmospheric Data Store

(https://ads.atmosphere.copernicus.eu/cdsapp#!/dataset/cams-global-reanalysis-eac4; Inness et al. 2019, last accessed November 2022). The sea ice data were obtained from the National Snow and Ice Data Center (https://nsidc.org/data/nsidc-0051/versions/2; DiGirolamo et al. 2022, last accessed October 2022), as were the surface data types (https://nsidc.org/data/g02156/versions/1, U.S. National Ice Center 2008, last accessed September 2023).

*Author contributions.* Both authors played a role in the study's design and deciphering the results. RJMW conducted the analysis and drafted

the manuscript, while EG provided feedback and comments.

*Competing interests.* The authors declare that they have no conflict of interest.

*Acknowledgements.* RMW and EG were supported by funding from the Royal Society (University Research Fellowship URF/R1/191602).



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
