# Peer review of "Air mass history linked to the development of Arctic mixed-phase clouds"

_EGUsphere, 2024_

## Author Comment (AC1)

**Air mass history linked to the development of Arctic mixed-phase clouds**

Rebecca J. Murray-Watson[1] and Edward Gryspeerdt[1]

[1]Space and Atmospheric Physics Group, Department of Physics, Imperial College London, UK

**Correspondence:** Rebecca J. Murray-Watson (rebecca.murray-watson17@imperial.ac.uk)

We thank the reviewers for their care in reading our manuscript, all of their suggestions have been carefully considered and our responses are attached below. Please note that the line numbers referenced below when discussing where edits have been made correspond to the diff manuscript.

**1 Response to Reviewer 1**

**In my opinion, Section 2 (Methods) does not provide sufficient detail. I understand that this study uses methods from two other papers that are cited, but I think some basic information is not reported. For example, which years are considered in the analysis? Which region do you use to generate the forward trajectories? For the MODIS retrievals described in Section 2.3, you filter out quite a bit of data; what fraction of the data do you end up considering? What does the so-called heterogeneity index tell you and why filter using it? Moreover, I see that you later cite Khanal and Wang (2018), so you're aware of the study, but I wonder how confident you are that MODIS is properly classifying "single-layer, liquid clouds" in this rather challenging environment. It would be good to say something about that.**

We thank the reviewer for their comments, and have implemented the following changes based on their comments on the Methods Section:

1) The years considered have been added at line 141 (2008-2014, inclusive).

2) The region considered (North Atlantic and Kara Sea) was included at line 140.

3) The fraction of MODIS data remaining after filtering was included at line 196 (40%).

4) The following sentence was included at line 191 to explain the MODIS heterogeneity index and justify its use: The MODIS cloud subpixel heterogeneity index (SPI) is the ratio of the standard deviation to the mean of the reflectance in the 0.86 $\mu$m band of the 250 m pixels which comprise each 1 km; larger values of SPI (>0.3) are associated with heterogeneous cloud regimes, such as broken cumulus fields (Zhang and Platnick, 2011).

5) Our apologies - it should have been mentioned that the MODIS data were filtered to only include pixels which were colocated with the DARDAR data, which had been filtered to single-level clouds, etc. This has been included in line 187. However, previous comparison studies have found that both the MODIS optical phase and single-layer cloud filters typically compare well against active sensors, particularly over ocean (e.g., Marchant et al. 2016; Wang et al. 2016). These references have been added to the text and lines 190-191.

**L133: Please define "TSI".**

A definition has been added at line 132.

**L137-139: Presumably warm air intrusion clouds make up an important fraction of these "other" clouds when the flow is poleward?**

This is true - we would expect warm air intrusions to form a significant fraction of the air mass moving into the Arctic, particularly in winter and spring. However, as the "Towards" clouds only serve as a qualitative comparison to the MCAO clouds and to help identify retrieval biases, they are not specifically considered in this study. However, we include a mention of them in line 139 to highlight their contribution to these trajectories.

**L157: "The temperature data are obtained from ERA5 and processed as the wind data"; what does this mean?**

The sentence has been rephrased for clarity at line 159, and the line now reads: "The temperature data are obtained from ERA5 and processed to the same grid as the wind data."

**L198: I see an increase in liquid cloud fraction over time in the "Towards" clouds, not a shift toward mixed or ice phases. Am I missing something? Figure 1: Is the x-axis showing time since tracking the airmass? Also, what does the solid line and shading represent?**

Both of these comments are considered together as they require clarrification for the "Towards" clouds - the time axis for these clouds is the time *until* the air parcel reaches the ice edge, so essential is the reverse of the time since ice (consider it a time until ice). The caption of Figure 1 has been changed to emphasise this distinction:

Evolution of cloud phase as a function of time for (a) MCAO, (b) non-MCAO and (c) Towards trajectories. Note that for Towards clouds, the x-axis represents hours until the clouds reach the ice edge and their development is instead read from right to left. For MCAO and non-MCAO clouds, the x-axis represents time since the air parcel left the ice edge.

**Figures 2, 9, and 10: I think it should be possible to determine which bins are statistically significant and which are not. Please add hatching or something similar to indicate this.**

Dots have been added to Figures 2 and 10 to indicate the significant points based on the Mann-Whitney U test (Mann and Whitney, 1947) at the 95% confidence interval, and the captions adjusted appropriately. It is notable that no points in Figure 9 were found significant at this confidence level, probably due to the the relative lack of points. However, the smoothness of the field in area C still suggests that these data generally support the conclusion in the text. Reference to this is included in line 310.

**Section 4: You discuss secondary ice production and link it to previous studies that have examined the role of strong updrafts. Considering that MCAOs are largely surface-driven, the M value provides a good proxy for surface turbulent heat fluxes. I wonder if you could filter your data by M value (e.g., bin positive values by increments of 3 or 4) to address your hypothesis.**

We agree that one would expect higher amounts of secondary ice production in stronger outbreaks. However, splitting the data up into such bins introduces a significant amount of noise, making it difficult to drawn conclusions. However, we believe that dividing the data into MCAO and non-MCAO events serves to highlight the effects of the surface heat fluxes in the same

way as splitting into strong and weak events, and these data supports the qualitative argument made for this difference in phase fraction.

**L364: COMBLES –> COMBLE**

This has been amended in line 380.

**L364-366: Please also note that COMBLE was measuring CAO conditions 100s-1000 km downstream of the ice edge.**

This has been added at line 380.

**L445: We know that secondary ice production can be at play; I suggest changing the phrasing to something like, "if the mechanism is ubiquitous".**

Line 462 has been rephrased as suggested.

**2 Response to Reviewer 2**

**The meteorological context seems to be lacking. I believe providing the meteorological context (showing and contrasting temperature (not just the cloud top temp), humidity, dynamics) would be very useful for the potential readers. This would also help to indirectly evaluate if the role of changing INPs makes physically sense. I understand that this can be a manuscript on its own. So, I am certainly not expecting a detailed analysis, but I think at least a brief overview is warranted to fully grasp what is going on as the air masses are transported large distances off the sea ice edges.**

**Another thing that got me thinking is the use of AOD as a proxy of INPs here. Given the persistence of clouds, high solar zenith angles and challenging meteorological and surface conditions, it is to be honest difficult to believe AOD changes that are in the order of 0.01 based on CAMS reanalysis. It would be better if the authors show a bit more information on aerosol variability. For example, are the changes in AOD really consistent with the hypothesis along the trajectories?**

**Furthermore, it is to be noted that the atmospheric circulation and the aerosol vertical structure in itself (at the cloud base, free troposphere) could be different under non-MCAO and MCAO scenarios, which could complicate the use of AOD.**

We thank the reviewer for these insightful comments, and will consider them together.

We have included a panel on environmental conditions, shown in Figure 1. It can be seen that in general for MCAO events, the strength of the instability decreases, while the AOD, specific humidity and surface temperature all increase as the air masses move away from the ice edge, typically to lower latitudes. A new section (3.3) describing the conditions has also been included.

In particular, the increase in AOD along the trajectory is in agreement with the hypothesis put forward in the paper: if a certain (unknown) fraction of these aerosols are assumed to act as INP, then as these are transported to the cloud layer given the strong surface fluxes, ice is more likely to form in these outbreaks. We agree that AOD is an imperfect metric to use for INP, especially given the lack of vertical profile structure. Here, AOD should be thought of as an indicator of how clean the air

mass is and how long it has spent over the ice - this has been clarified in line 453. Given a lack of other information available in the region, here it is used to support the hypothesis presented in this work. Although other CAMS data could be used in this study (such as aerosol mass at different levels), AOD also allows for this work to be compared with retrieved AOD in future studies. As such, we continue with its use here.

A new section discussing the meteorological context has been added (Section 3.3 in the new manuscript), with references to the figure where relevant in the remainder of the manuscript.

[Figure]

**Figure 1.** The average evolution of (a) the MCAO index, (b) the AOD, (c) the surface temperature and (d) the specific humidity for MCAO and non-MCAO events. The shading represents the 95% confidence interval.

**3   Response to Xinyi Huang**

**(1). The Lagrangian trajectories were generated by pixels being advected with 1000 hPa wind field. As cloud formation and development also include vertical air motion, with rapid changes in cloud phase on these timescales, how representative are the trajectories of indicating the air mass history involved in both the MCAO and no-MCAO clouds?**

We thank Xinyi Huang for their comments and interest in the paper. Several previous studies have considered the use of 1000 hPa to track low-level cloud in various settings (e.g., Gryspeerdt et al. 2022) and found that it is able to represent the advection well. In particular, these trajectories were used in Murray-Watson et al. 2023. Different pressure levels, up to 800 hPa, were tested for the trajectories with minimal differences to the results.

**(2). How important are other cloud microphysics processes to the change and evolution of the cloud phase? For example, the removal of liquid water by autoconversion and accretion which could also potentially increase the ice and mixed-phase fractions during the lifetime of MCAO clouds. Do the authors have any evidence to eliminate these alternative explanations?**

Once ice is present, ice-related processes typically dominate the loss of water from the cloud through mechanisms such as the WBF process or the riming of liquid droplets onto ice particles. Therefore, we do not expect liquid-phase processes such as autoconversion or accretion to contribute to affect the evolution of the cloud phase.

**References**

115 Gryspeerdt, E., Glassmeier, F., Feingold, G., Hoffmann, F., and Murray-Watson, R. J.: Observing short-timescale cloud development to constrain aerosol–cloud interactions, Atmospheric Chemistry and Physics, 22, 11 727–11 738, https://doi.org/10.5194/acp-22-11727-2022, 2022.

Mann, H. B. and Whitney, D. R.: On a Test of Whether one of Two Random Variables is Stochastically Larger than the Other, The Annals of Mathematical Statistics, 18, 50 – 60, https://doi.org/10.1214/aoms/1177730491, 1947.

120 Marchant, B., Platnick, S., Meyer, K., Arnold, G. T., and Riedi, J.: MODIS Collection 6 shortwave-derived cloud phase classification algorithm and comparisons with CALIOP, Atmospheric Measurement Techniques, 9, 1587–1599, https://doi.org/10.5194/amt-9-1587-2016, 2016.

Murray-Watson, R. J., Gryspeerdt, E., and Goren, T.: Investigating the development of clouds within marine cold-air outbreaks, Atmospheric Chemistry and Physics, 23, 9365–9383, https://doi.org/10.5194/acp-23-9365-2023, 2023.

125 Wang, T., Fetzer, E. J., Wong, S., Kahn, B. H., and Yue, Q.: Validation of MODIS cloud mask and multilayer flag using CloudSat-CALIPSO cloud profiles and a cross-reference of their cloud classifications, Journal of Geophysical Research: Atmospheres, 121, 11,620–11,635, https://doi.org/https://doi.org/10.1002/2016JD025239, 2016.

Zhang, Z. and Platnick, S.: An assessment of differences between cloud effective particle radius retrievals for marine water clouds from three MODIS spectral bands, Journal of Geophysical Research: Atmospheres, 116, https://doi.org/10.1029/2011JD016216, 2011.